# Autism Spectrum Disorder Pathogenesis—A Cross-Sectional Literature Review Emphasizing Molecular Aspects

**DOI:** 10.3390/ijms252011283

**Published:** 2024-10-20

**Authors:** Agata Horecka-Lewitowicz, Wojciech Lewitowicz, Monika Wawszczak-Kasza, Hyebin Lim, Piotr Lewitowicz

**Affiliations:** 1Institute of Medical Sciences, Jan Kochanowski University, Al. IX Wiekow Kielc 19A, 25-516 Kielce, Poland; 2Student Scientific Society at Collegium Medicum, Jan Kochanowski University, Al. IX Wiekow Kielc 19A, 25-516 Kielce, Poland; wojciech.lewitowicz@ujk.edu.pl (W.L.); hyebinlim@yahoo.com (H.L.); 3Institute of Health Sciences, Jan Kochanowski University, Al. IX Wiekow Kielc 19A, 25-516 Kielce, Poland

**Keywords:** ASD, Asperger’s syndrome, autism, molecular sequencing, neurodevelopment, synapses, ion channels, etiology

## Abstract

The etiology of autism spectrum disorder (ASD) has not yet been completely elucidated. Through time, multiple attempts have been made to uncover the causes of ASD. Different theories have been proposed, such as being caused by alterations in the gut–brain axis with an emphasis on gut dysbiosis, post-vaccine complications, and genetic or even autoimmune causes. In this review, we present data covering the main streams that focus on ASD etiology. Data collection occurred in many countries covering ethnically diverse subjects. Moreover, we aimed to show how the progress in genetic techniques influences the explanation of medical White Papers in the ASD area. There is no single evidence-based pathway that results in symptoms of ASD. Patient management has constantly only been symptomatic, and there is no ASD screening apart from symptom-based diagnosis and parent-mediated interventions. Multigene sequencing or epigenetic alterations hold promise in solving the disjointed molecular puzzle. Further research is needed, especially in the field of biogenetics and metabolomic aspects, because young children constitute the patient group most affected by ASD. In summary, to date, molecular research has confirmed multigene dysfunction as the causative factor of ASD, the multigene model with metabolomic influence would explain the heterogeneity in ASD, and it is proposed that ion channel dysfunction could play a core role in ASD pathogenesis.

## 1. Introduction

Autism spectrum disorder (ASD) was first described by Leo Kanner in 1943 [1]. In the current medical nomenclature, it covers autism, atypical autism, Asperger’s syndrome, and a pervasive developmental disorder not otherwise specified. Male patients are much more frequently affected than female patients.

ASD is manifested by constant deficits in social communication and social interaction, as well as by restricted, repetitive behavior, interests, or activities. The mechanism of ASD remains unclear. It is currently suspected to have a multilineage etiology. The increasing incidence of these disorders represents a significant medical problem associated with a surge in both medical and social care costs. Currently, the progress that has been made regarding our understanding of the genetic basis of ASD has improved comprehension and screening of ASD risk. ASD prevalence has been measured by special education and administrative records [2,3,4,5].

Generally, the frequency of ASD occurrence in children is estimated to be approximately 1.5–2%. A cross-sectional study by [Christiansen et al]. covering the years 2010, 2012, and 2014 showed that the estimated prevalence of ASD (per 1000) among children aged 4 years was 13.4 in 2010, 15.3 in 2012, and 17.0 in 2014 [2]. These results indicate a rising incidence of ASD. However, typical clinical symptomatology is presented the symptoms severity vary in many cases. The milder form was referred to in the past as high-functioning Autism (HF ASD) [6].

The diagnosis of ASD is still only symptomatic and is based on DSM-5 criteria [7]. There is still no biomarker which would be helpful in the screening and surveillance of ASD-affected patients. However, the dropping cost of molecular tests gives hope for wide testing, but clinicians and parents have yet to benefit from this. Patient management is also still only symptomatic and ineffective in most cases. There is still no causative approach or targeting therapy.

The families of ASD patients were left to provide care with no guidance or advice for many years, which resulted in some forming stereotypical and misinformed speculations or beliefs, some of which are unfortunately still present today.

The aim of this review is to show how approaches explaining ASD etiology have been evaluated in recent decades. The objective of this review is to show how far we have come in relation to neuroscience, especially with a focus on ASD. In this paper, we present the mainstream theories in ASD pathogenesis.

## 2. Materials and Methods

### Searching Methodology

We pointed to execute a narrative review covering the core streams in ASD explanation. All the reviews in this paper, ranging from 1998 to 2024, were obtained by means of a systematic search of Pubmed, Scopus, and the Web of Science. Conducted searches included the following keywords: autism, ASD, Asperger’s syndrome, genetic profile, gut–brain axis, food intolerance, microbiota, cell migration, and brain development. The exclusion criteria were animal studies and case reports. As an inclusion criterion, we included research based on humans in both original papers and reviews. Moreover, we included studies with the following characteristics:Peer reviewed;Availability of the full-text publication;Availability in English.

The reviewed studies contained demographically diverse samples originating from a wide range of Western cultural contexts.

## 3. Results

The collected studies allowed us to present a multilineage approach to explain ASD pathogenesis. Everything we know about ASD to date has been provided by research conducted in Western high-income countries. It is believed that most ASD patients live in low- and middle-income regions, where access to modern medicine is still heavily restricted. There are multicultural differences in behavioral norms, culture-specific approaches, beliefs, mental health literacy, or even stigmas [8]. To the present day, the etiology of ASD is still enigmatic. Many researchers have tried to explain its driving abnormality, but there is still no hard evidence. From a time perspective, there is an evident change in perception of ASD etiology. The first highlighted idea was the gut–brain axis theory with an emphasis on gut microbiota, especially the fungal colonization of the colonic mucosa. A parallel theory centered around the influence of environmental changes, especially environmental pollution. Subsequently, the advancements made in laboratory technology allowed for studies that focus on stem cell pathways and molecular abnormalities.

The problem overview—from the past to present day

### 3.1. Gut–Brain Axis

The relationship between the gut environment and the brain (microbiome–gut–brain axis) has been seen as an important factor in modulating brain maturation and functioning, but its detailed contribution to ASD etiology is not well understood [9].

The discussion about the ASD pathway has focused on the gut–brain axis for many years. Many studies presented a dependence of ASD symptoms on gut dysbiosis. Generally, intestinal dysbiosis and frequent gastrointestinal (GI) symptoms partially explain the role of probiotics and other para-pharmaceuticals in the care of children. The microbiota composition built by colonies of Bacteroidetes, Firmicutes, and Actinobacteria is much more concentrated in children with ASD than in controls. ASD patients showed an abnormally high ratio of Faecalibacterium and Phascolarctobacterium and fewer Coproccocus and Bifidobacterium species. Assuming the presence of several bacterial, fungal, and viral species in the gut microbiota, there is a risk of imbalance between commensal and colonic mucosa which can lead to the relaxation of intracellular connections and finally to leaky gut. The alternation between neural, endocrine, and metabolic mechanisms is influenced by the metabolome and could be crucial in gut–CNS signaling. The authors concluded that the administration of probiotics would be the most promising treatment for neurobehavioral symptoms; however, there is still a need for well-designed trials in this field [10,11,12,13,14,15]. GI symptoms are four to five times more frequent in ASD patients compared to the normal population. A spectrum of symptoms was found, such as abdominal discomfort, constipation alternating with diarrhea, bloating, esophageal reflux, vomiting, and food allergies. An altered metabolism and absorption of disaccharides is observed in the gut epithelium. Autistic children present with disaccharidase malabsorption due to decreased levels of lactase, maltase, and glucoamylase. Consequently, a high concentration of sugars leads to fermentation, an altered microbial composition, osmotic diarrhea, and bloating. It is believed that bidirectional communication between the gut and the brain is via vagal fibers and gut neurotransmitters. Leaky gut and coinfection with Clostridiaceae spp. lead to the production of toxic metabolites such as phenols, p-cresol, and indoles. P-cresol presents some hope because it could be a possible biomarker for ASD due to its elevated levels in the urinary and fecal samples of autistic children. P-cresol inhibits dopamine-beta-hydroxylase and could modify dopamine levels in the brain. A wide biochemical study revealed other colonic microbiota metabolites which could be involved in ASD pathogenesis, such as tryptophan, glutathione, homocysteine, methionine, and short-chain fatty acids [11].

In contrast, a recent systematic review by Ho et al. (2020) discredits gut dysbiosis as a leading cause of ASD symptoms. According to the authors, the conclusions in reports are ambiguous and so their causality cannot be confirmed [16].

Another proposed aspect of gut dysfunction is diet intolerance. It is believed that a gut–brain axis dysfunction could lead to colonic smoldering with secretion many cytokines influencing to neural cells. Moreover, the inflammatory pathway is observed also in other neurogenerative diseases [17,18].

### 3.2. Role of the Environment

A few studies investigating other ASD causes in diet or environmental associations focus on mercury and biphenyl poisoning as a potential ASD cause. A review by [McCaulley (2019)] suggests the neurotoxicity of a common heavy metal via genetic and epigenetic alterations [19]. The fact the industrial progress has led to severe environmental pollution is non-negotiable. In the list of the most harmful chemicals, there are polychlorinated biphenyls (PCBs) and heavy metals with their proven neurotoxicity. The reported research highlights the evidence of PCBs alternating with dopamine (DA) neurotransmission. In addition, in vitro studies using pheochromocytoma cell lines exposed to PCBs presented a significant reduction in dopamine concentrations [20,21,22]. Furthermore, the molecular mechanism underlying reduced cognition, attention, behavior, attention deficit, hyperactivity disorder, or ASD has also been highlighted. All these have been found to impact dopaminergic neurotransmission, hypothyroidism, calcium dyshomeostasis, or oxidative stress, especially in early childhood, presenting ASD-specific symptoms [19]. Recently, it has been reported that high levels of reactive oxygen species (ROS) leading to oxidative stress could also be a possible mechanism resulting in ASD. This pro-inflammatory state caused by ROS results in dysregulation of the central nervous system and immune system, which has been acknowledged as contributing factors of ASD development. Moreover, it is thought that reduced resources of free-radical scavengers such as decreased glutathione levels make individuals more vulnerable to neuron injury.

The signaling pathway of lipid molecules contributes to the pathophysiology of autism spectrum disorder (ASD) and provides hope for alternative therapeutic strategies.

Any form of cell membrane injury leads to cyclooxygenase-2 activation and the secretion of pro-inflammatory cytokine. Prostaglandin E2 as a membrane-derived lipid molecule contributes to ASD with canonical Wnt signaling. The endocannabinoid system maintains a balance between inflammatory and redox status and synaptic plasticity and is a potential target for an ASD pathway. There is believed that redox signaling pathway and redox-status-sensitive transcription would play an important role as cofactors in ASD genesis. Cannabinoids can modulate the redox status by antioxidant molecules enhancement via ROS-producing NADPH oxidase (NOX) and superoxide dismutase [23]. Lagod et al. reported that elevated levels of short-chain fatty acids (SCFAs), especially propionic acid (PPA) due to gut dysbiosis, contribute to ASD [24].

Any type of maternal immune system activation via secreted cytokines can impact on the fetal CNS environment. The activation of monocytes, macrophages, mast cells, and microglia and the high production of pro-inflammatory cytokines are indeed the causes of smoldering neuroinflammation, and the latter disturb glia–microglia homeostasis. Many researchers put attention on the mitochondrion as crucial place to start inflammation [25,26,27,28,29].

Another metabolic route concerning L-carnitine deficiency was considered a unique subgroup of ASD patients based on the suggestion of mitochondrial dysfunction, leading to the first autistic symptoms. Following that path, the recently reported clinical trials conclude that carnitine administration could mute severe symptoms in non-syndromic ASD. Unfortunately, in both trials, dose-dependent adverse reactions were observed, but the same beneficial effect has also been reported for other comorbid disorders, such as intellectual disability and increased muscle tension. The authors concluded that more studies are necessary; however, the beneficial features of carnitine treatment based on mitochondrial dysfunction may alleviate the symptoms in non-syndromic ASD [30].

### 3.3. Stem Cells

Neural stem cells (NSCs) are present in the central nervous system and can differentiate into neurons, astrocytes, and oligodendrocytes. The behavior of these cells is regulated by various factors, determining whether they remain quiescent or functionally active. CNS maturation is based on correct signaling by many neurotransmitters. These newly formed neurons contribute significantly to the plasticity and functionality of the nervous system. Because ASD etiology is joined with altered neural maturation, this aspect is also the focus of ASD research.

The innovative thesis presented by Bankaitis et al. concerning the impact of L-carnitine was discussed last year. The authors coined the neural stem cell (NSC)/carnitine malnutrition hypothesis. According to the authors, an ASD risk factor could be the diminished capacity for carnitine-dependent long-chain fatty acid β-oxidation in the neural stem cells of the developing central nervous system. The authors concluded that fetal carnitine status is a significant metabolic component in determining NSC vulnerability and could further contribute to abnormal cell maturation and dysfunction [30,31].

With obvious reason, the study of glial cells in ASD-affected patients is impossible. The hope is that cellular models can provide us with a way to uncover disease mechanisms and develop novel therapeutic strategies. The ability of induced pluripotent stem cells (iPSCs) to generate diverse brain cell types offers great opportunities to study neurodevelopmental disorders.

This iPSC-based model was used to understand the neuronal and glial contributions to neurodevelopmental disorders, including ASD, Rett syndrome, bipolar disorder (BP), and schizophrenia. For example, many molecular hotspots have been shown to influence cellular phenotypes in three-dimensional iPSC-based models in patients with ASD. Delays in the differentiation of astrocytes and morphological changes in neurons are associated with Rett syndrome. In bipolar disease and schizophrenia, patient-derived models helped identify cellular phenotypes associated with neuronal deficits and mutation-specific abnormalities in oligodendrocytes [32].

### 3.4. Still-Scattered Genetic Puzzle

The last three decades encompasses a period of significant progress in genetic studies concerning neurobehavioral syndromes including ASD. The first results provided a significant amount of molecular data that were used to verify many hypotheses, models, and, finally, genetic pathways of ASD. To the present day, several lines of evidence support the view that structural and genomic variation play a pivotal role in ASD. All sophisticated molecular techniques for genome sequencing, including array-comparative genomic hybridization and single-nucleotide-polymorphism analysis, have allowed for the detection of a large number of autism-related loci. Copy number variants (CNVs) are the most common form of molecular abnormality and are seen as very important contributors to the pathogenesis of neurodevelopmental disorders. Because of the complex synaptic architecture, deciphering the functional impact of ASD-associated variants is an extremely arduous task. Currently, it is believed that at least hundreds of loci are associated with ASD. That complexity makes it difficult to identify singular potential causative pathways and then therapeutic approaches. Analyzing the databases, we can find a significant amount of research based on diversified genetic material and used molecular assays. That is why we are lagging considerably in interpreting the main causative molecular sequences in ASD. The system biology/network analysis approaches provided new insights into the molecular driving pathway. To help with these analyses, animal models, in vitro studies, and experimental approaches contributed as well. It is suggested that a comparison of these data on a multilevel plane would allow an ASD template model to be created and then both psychiatric dysfunction and other somatic problems to be explained [33].

Figure 1 presents the main causative factors contributing to ASD genesis.

Previously mentioned CNVs are particularly important in the cases of complex syndromes, such as when ASD symptoms occur in association with intellectual disability and/or congenital malformations (e.g., Angelman syndrome, Phelan–McDermid syndrome [33]).

The model proposed by Bourgeron to explain the complex genetic landscape in ASD appears to be very reasonable. Because of the massive molecular heterogeneity in affected subjects, the authors try to focus on the interplay between a genetic background and a low- to high-risk predisposition to ASD onset. Moreover, there are new rare or ultra-rare variants with low- to high-risk potential which can contribute to ASD as well. The combination of these different categories of variants in the population results in the massive phenotypic heterogeneity of ASD. Reported data provided evidence concomitant with different models of inheritance in the heterogeneous ASD population, but there is still a lack of a driving core. Here, we can find a monogenic presence in subjects carrying ultra-rare or de novo variants of extremely deleterious and highly penetrating mutations; an oligogenic presence with the concomitance of medium/high-risk predisposing variants; and, finally, a polygenic presence of multiple low-risk genetic variants [34]. This subtyping would explain clinical ASD types and symptom diversity and severity. During the last two decades, the launch of high-throughput sequencing revolutionized genetic research and allowed ASD to be studied on a significantly wider molecular landscape. The early documentation on karyotype chromosomal abnormalities shed light on susceptible loci screening in those highly involved genomic regions such as chromosomes 7q, 1p, 3q, 16p, and 15q [35]. Sequencing technology undoubtedly confirmed that the etiology of ASD is multigenic and highly heterogeneous. At this moment, it is known that any ASD patient bears various CNVs, raising ASD susceptibility and providing additional proof of the multigenic etiology. Although there is no clearly defined biomarker or driving molecular route identified, fresh research on DNA methylation presented other genes involved in ASD. Only a small handful of ASD-related diseases have a monogenic cause, such as Rett syndrome, tuberous sclerosis, fragile X syndrome, and Schuurs–Hoeijmakers syndrome [35,36,37,38,39]. These very well signify the Bourgeron hypothesis.

The complexity of neurodevelopment and signal transduction makes multilevel neurocyte dysfunction possible. Genes output as a functional protein contribute in synapse formation, transcription regulation, and even chromatin remodeling.

In this group, synapse-related risk genes were included, especially cell-adhesion proteins and ion channels. The synapsis family of proteins contribute to synaptogenesis and the release of neurotransmitters. Mutations in synapsin-1 (SYN1) and synapsin-2 100 (SYN2) genes are common in neuropsychiatric disorders including ASD. Another problem observed in ASD is the changed signal transduction and influence on neurotransmitter secretion. The reported data highlighted a high ratio of mutations in genes encoding ion channel protein such as sodium voltage-gated channel alpha subunit 2 (SCN2A); potassium voltage-gated channel subfamily D member 2 (KCND2); calcium voltage-gated channel subunit alpha1 E (CACNA1E); and Ankyrin, which is a protein encoded by multiple ankyrin repeat domains 3 (SHANK3), which play important roles in synaptogenesis, the maintenance of membrane channels, and the clearing of synaptic cleft [40,41,42,43,44,45,46]. Figure 1 presents overlapping of ASD causes. Clinical observation and key ASD symptoms confirm an implication of synapse dysfunction and abnormal dendritic networks in ASD. A wide gene panel sequencing allowed an increased number of de novo mutations (DNMs) to be found in regulatory genes. However, a correlation of DNM elements within the targeted genes on neuropsychiatric disorders was not identified yet, and the DNMs were found to be specifically upregulated in early prenatal brain development [47,48,49,50]. Another no-less-significant aspect in ASD genesis is chromatin-remodeling pathways. Methyl CpG binding protein 2 (MeCP2) working as an activator or inhibitor of other genes is included in this gene group. Ubiquitin Protein Ligase E3A (UBE3A) and chromodomain helicase DNA binding protein 8 (CHD8) are enzymes contributing to the proteasome degradation of proteins via connection with ubiquitin. Mechanically, UBE3A silencing could be activated by methylation, confirming what was proven with Angelman and Willi–Prader syndrome. Tran and co-workers recently showed that fragile X mental retardation protein (FMRP) and fragile X-related protein 1 (FXRP1) mutations could lead to abnormal RNA editing enzyme activity, leading to the hypo-edition of adenosine–inosine transformation in neurons [51,52].

It is thought that disease-causing gene mutations are germinal and are present in almost all somatic cells. Obviously, post-zygotic acquired mutation could lead to a somatic mosaicism, which is common in neurodevelopmental diseases, including autism. Neurogenesis is a crucial period in tissue development and maturation and acquired SNVs could lead to gene polymorphism. It was especially observed in the example of the sodium channel alpha-1 subunit, SCN1A [53,54,55]. Generally, according to much reported data, the acquired mutation prevalence is calculated to be approximately 5–7%. However, the frequency of post-zygotic mutation in autism is unknown, and some research highlights its role as an important pathway in ASD genesis [56,57]. Most reported mutations are not pathogenic or are classified as unknown meaning, but some exon polymorphisms could be extremely detrimental. All of this has been associated with ASD, Rett syndrome, tuberous sclerosis, and intellectual disability. Before next-generation sequencing was performed, our understanding of somatic mosaicism in ASD was created only by simple molecular tests and were reported as case reports. The whole-exome sequencing (WES) data from large cohorts have been a milestone in understanding the role of somatic mosaicism. According to [Krupp et al. and Lim et al.], the mosaicism prevalence was estimated to be 3–5%. Other research based on a large cohort covering 5947 families affected by ASD studied the meaning of critical exon variations which were much more common in ASD children than in unaffected siblings. In addition, the authors pointed out that these exon variants showed higher expression in the amygdala—an area critical for emotional response and social awareness [58]. A similar large-cohort WES study by Freed and colleagues reported similar conclusions concerning somatic mosaicism as a significant factor in ASD etiology [59].

Currently, CNVs are now seen as a critical driving factor in ASD susceptibility. Additionally, there is a thesis that these variations directly cause approximately 10% of all ASD cases. The molecular tests pointed out 16p11.2 duplications as a very important region of the DNA chain. At least 25 genes involved in neurodevelopment and maturation are located in this region. [Golzio et al.] hypothesized that only one gene in this region—potassium channel tetramerization domain containing 13 (KCTD13)—is a pivotal driver for neuropsychiatric disease [60,61]. This strand has undergone further investigation. It was observed that CNVs in the 16p11.2 region undergo incorrect synaptic transmission through an altered regulation of Ras homolog family member A (RhoA) [62]. In addition, [Escamilla et al]. concluded that KCTD13 deletions are crucial in ASD genesis, but in contrast, [Golzio] concluded that this mutation alone is not likely to be sufficient to cause the disease. There is no single driver of disease in 16p11.2 region. Duplications or deletions is not from just one gene but from an interaction of all 25 genes contributing to ASD susceptibility. [Iyer et al.] systematically investigated the interaction between genes in the 16p11.2 region, using RNAi in Drosophila to test 565 pairwise knockdowns. Moreover, they presented 24 interactions between pairs of genes within the 16p11.2 region, as well as 46 interactions between 16p11.2 genes and other ASD-related genes. This information would be crucial in searching for a leading pathway in ASD genesis. Data suggest that interactions within CNVs result in ASD [63]. The other CNV loci are less frequently studied. The most frequent studied ASD-related regions, such as 15q11–13 and 16p11.2, are present in approximately 1% of cases [64].

Working with heterogeneity, we can see that paying attention to commonly affected functional networks proves to be a useful method of study. It is a prevalent outcome in numerous studies that autistic people have deletions in synaptic genes, namely SHANK3, dipeptidyl peptidase-like 10 (DPP10), neuroligins, and neurexins [65,66,67]. Among gene sets with rare CNVs, it is usual to see those related to cell development and proliferation, chromatin regulation, and ubiquitin pathways. In cases of some CNVs, copy number dosage seems to have an influence on disease phenotype. [Stefansson et al.] looked at the 15q11.2 CNV region of autistic people and discovered that there were two areas of the brain with dose-dependent structural and functional effects [68]. It is interesting to see that non-ASD/schizophrenic, dyslexic, and dystaxic controls manifested the very same structural changes. [Girirajan et al. (2010)] found a dose-dependent effect based on microarray analysis with identified CNVs in genes associated with ASD. The reported correlation was between a duplication size increase and autism severity. However, no relation was found between duplication size and nonverbal IQ [69]. The question to ask here is why do non-causative modifiers play a part in modulating CNV pathogenicity? Epigenetic gene-modulating functions have a great deal of involvement in ASD. There was a study which proposed that highly penetrative ASD-risk-related genes were usually located in the nucleus and that they have an involvement in the modulation of either expression or silencing of the protein–protein network, key for CNS development [70]. The studies present that the deep of epigenetic abnormalities could modify clinical symptoms. The study covering 50 pairs of monozygotic twins discordant of ASD who were reported to have many cases of autism associated with differentially methylated regions, with some patterns of CpG sites in line with symptom groups. Even though there remains a great amount of knowledge to be grasped about the epigenetic modulation of ASD, large-scale epigenomic studies have already provided the scientific and medical communities with valuable patterns. It is believed that the methylation of KMT2C, lysine methyltransferase 5-6B, MeCP2, CHD8, POGZ, FMRP, the RBFOX family, UBE3A, and E3 ubiquitin-protein ligase 1 would seriously improve ASD susceptibility [71]. These proteins are varied function wise and often include pathways seen in autism—for example, synaptic formation. To see how single epigenetic regulator mutations could account for the modification of numerous other risk genes, we could focus on the two leading susceptibility genes, namely MeCP2 and UBE3A. MeCP2 is a chromatin modifier which is, with no doubt, involved in ASD. It is seen in a case of a healthy individual that the binding action of MeCP2 regulates numerous synaptic function genes, GABRB3, brain-derived neurotrophic factor (BDNF), distal-less homeobox 5 (DLX5), insulin-like growth factor-binding protein 3 (IGFBP3), cyclin-dependent kinase-like 1 (CDKL1), protocadherin beta 1 (PCDHB1), protocadherin 7 (PCDH7), and lin-7 homolog A (LIN7A) [72,73,74]. The E3 Ubiquitin Protein Ligase UBE3A is the second crucial epigenetic regulator closely related to ASD. It strictly cooperates with MeCP2 and also could be causative itself.

UBE3A’s location is 15q11–13, regularly duplicated in autism cases. Dose-dependent effects correlated positively with the decreased excitatory synaptic transmission speech delay and psychomotor regression [75,76]. [Lee et al. (2014)] identified four proteasome-related UBE3A direct substrate proteins. UBE3A and substrate proteasome 26S subunit, non-ATPase 4 (Rpn10), caused the growth in accumulation of ubiquitinated proteins, hinting at a photostatic imbalance. Proteasome health is strongly implicated in dendritic spine outgrowth, making for a connection between UBE3A and abnormal neurocytes as seen in autism cases [77,78,79]. Additionally, its involvement in Wnt signaling could also cause a serious perturbation during the development period. MeCP2 and UBE3A are two fine examples of a gene mutation causing very extensive effects. The wide-ranging epigenetic studies provided us with a broad view of epigenetic dysregulation seen in ASD. Ladd-[Acosta et al. (2014)] measured more than 485,000 CpG loci in the postmortem brain tissue of 40 individuals, finding four differentially methylated regions. Three methylated regions were found in the cortex: the proline-rich transmembrane protein 1 3′ UTR, promoter regions of tetraspanin 32, and C11orf21. The other site was identified in the cerebellar, an alternative promoter for succinate dehydrogenase complex flavoprotein subunit A pseudogene 3 (SDHAP3) [80]. The evidence of abnormal DNA methylation in ASD cases exists in numerous aspects, ranging from genetic mutations in epigenetic machinery to loci-specific and genome-wide changes. It is possible for epimutations in DNA methylation to be acquired, and methylation reprogramming and imprinting are active during early embryogenesis and postnatal peak synaptogenesis [81]. [Mor et al.] approached this matter using small-RNA sequencing data, relating the results to genome-wide DNA methylation to find dysregulated miRNAs. This result proved to be in line with many other studies, and the significantly expressed miRNAs in the brains of ASD patients were related to synaptic activity. There was also mention of a connection to the oxytocin receptor OXTR gene, which hints at attenuated OXTR expression in the autistic brain. 

In Table 1, we presented a characteristic the common genes affected in ASD. In Clinically significant variants section, we summarize the variants reported in UniProt, ClinVar, LOVD, MitoMap, VarSome and PubMed according to its classification and coding impact. Worth of note is diversity of molecular alteration leading to ASD symptoms (access date 12 October 2024).

## 4. Conclusions and Future Perspectives

This review increases knowledge concerning ASD and presents our progress to the present day. However, there is no well-defined ASD causative factor; further challenges are ahead. There is no single evidence-based pathway that results in symptoms of ASD. Patient management has constantly only been symptomatic, and there is no ASD screening apart from symptom-based diagnosis and parent-mediated interventions. Multigene sequencing or epigenetic alterations hold promise in solving the disjointed molecular puzzle. Further research is needed, especially in the field of biogenetics and metabolomic aspects, because young children constitute the patient group most affected by ASD.

A novel approach concerning neuroinflammation as an important contributor in ASD genesis sheds new light on targeting therapy. There is pioneering research focusing on exosomes. These exosomes are subcellular nanoparticles able to transfer various types to messengers such as fragments of DNA, RNA, hormones, cytokines, and other proteins which can be transmitted across the blood–brain barrier. Because of progress in bioengineering, there is an idea to use the exosomes harboring anti-inflammatory molecules. This solution could regress ASD symptoms such as irritability, aggression, and socialization difficulties.

## Figures and Tables

**Figure 1 ijms-25-11283-f001:**
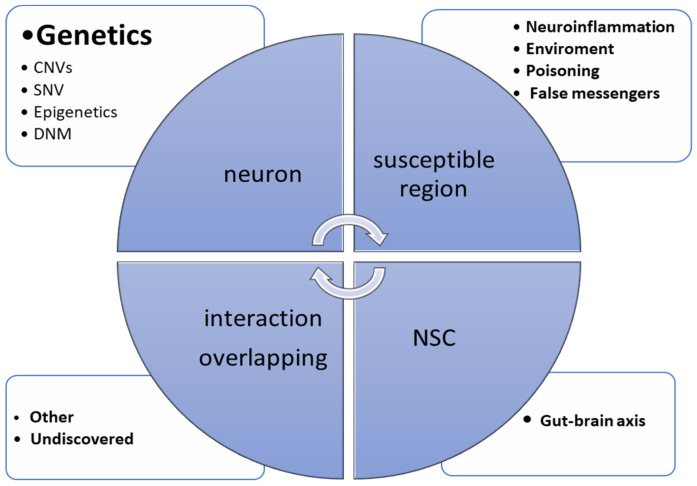
Chart presenting the most important pathways involved in ASD. Moreover, it presents overlapping and interaction the most common malfunctions observed in ASD. (CNVs—copy number variations; SNV—single-nucleotide variations; DNM—de novo mutation; NSC—neural stem cell).

**Table 1 ijms-25-11283-t001:** The most common genes involved in ASD, their location, pathogenic variants, and function.

Gene	Variations	Clinically Pathogenic Significant Variants (No. of Cases)	Locus	Function	Clinical Consequences of Alteration
SYN1 SYN2	Deletion, duplication, SNV	Missense (6) Nonsense (12) Frameshift (35) Inframe Indel (1) Splice junction loss (8) -	chrX:47,571,901–47,619,857 and chr3:12,004,388–12,192,032	Synapse formation	X-linked delayed intellectual development, X-linked epilepsy, Pallister–Hall syndrome, schizophrenia, ASD, epilepsy
KCND2	CNV	Missense (7)	chr7:120,272,908–120,750,337	Potassium voltage-gated channel, mediates transmembrane potassium transport in the brain	Early Myoclonic Encephalopathy
CACNA1E	CNV	Missense (31) Nonsense (5) Frameshift (2) Splice junction loss (1)	chr1:181,317,690–181,813,262	Voltage-dependent calcium channel alpha 1e subunit; involved in the modulation of calcium-mediated hormone secretion	Spastic tetraplegia, cerebral cortical atrophy, Developmental regression
SHANK3	Deletion methylation	Synonymous (1) Missense (23) Nonsense (38) Frameshift (135) Splice junction loss (12) Non-coding (1)	chr22:50,672,823–50,733,212	Multidomain scaffold proteins of the postsynaptic density that connect neurotransmitter receptors, ion channels, and other membrane proteins to the actin cytoskeleton and G-protein-coupled signaling pathways. Also plays a role in synapse formation and dendritic spine maturation	Social communication, behavior, language, synaptic transmission, motor delay, delayed CNS myelination, seizures
MecP2	Methylation	Synonymous (15) Missense (168) Nonsense (87) Start loss (8) Stop loss (20) Frameshift (482) Inframe Indel (41) Splice junction loss (28) Non-coding (2)	chrX:154,021,573–154,137,103	Chromosomal protein that binds to methylated DNA the corepressor SIN3A	Cooperation with other genes, repressor synapse development Severe neonatal onset encephalopathy with microcephaly
UBE3A	Methylation	Synonymous (1) Missense (52) Nonsense (66) Start loss (1) Stop loss (3) Frameshift (122) Inframe Indel (11) Splice junction loss (13) Non-coding (2)	chr15:25,333,728–25,439,056	Encodes an E3 ubiquitin-protein ligase, part of the ubiquitin protein degradation system	Loss of imprinting, motor delay, intellectual disability, delayed speech and language development
KCTD13	Deletion duplication	-	chr16:29,905,012–29,926,236	Contributes to ubiquitin-protein transferase activity.	Reduced NSC proliferation and maturation, ASD, Intellectual Disability Syndrome
KMT2C	CNV	Synonymous (1) Missense (17) Nonsense (54) Frameshift (69) Inframe Indel (1) Splice junction loss (15) Non-coding (2)	chr7:152,134,922–152,436,644	Possesses histone methylation activity and is involved in transcriptional coactivation	Cerebellar hypoplasia, motor delay, delayed speech and language development, global development delay, seizures
GABRB3	CNV	Missense (65) Nonsense (7) Frameshift (3) Inframe Indel (3) Splice junction loss (4)	chr15:26,543,552–26,939,539	Oncoprotein acts as multi-subunit chloride channel that serves as the receptor for gamma-aminobutyric acid, a major inhibitory neurotransmitter	Epilepsy, childhood absence epilepsy, Lennox–Gastaut Syndrome
BDNF	CNV	Missense (1) Nonsense (2)	chr11:27,654,893–27,722,058	Encodes a nerve growth factor family of proteins. During development, promotes the survival and differentiation of selected neuronal populations of the peripheral and central nervous systems.Participates in axonal growth, pathfinding and in the modulation of dendritic growth and maturation	Cognitive impairment, seizure, intellectual disability, neuroblastoma, ganglioneuroblastoma
PCDDH7, PCDHB1	CNV	-	chr5:141,051,374–141,059,346	Potential calcium-dependent cell-adhesion protein.May be involved in the establishment and maintenance of specific neuronal connections in the brain	Rett syndrome, schizophrenia, ASD
FOXP2	Deletion	Missense (4) Nonsense (27) Frameshift (13) Splice junction loss (6)	7q.31.1	Plays role in developing neural, gastrointestinal and cardiovascular tissues.Plays a role in synapse formation by regulating SRPX2 levels.Involved in neural mechanisms mediating the development of speech and language	Mental retardation, many organ development Delayed speech and language development Abnormal basal ganglia morphology Deficit in grammar Skeletal muscle atrophy
IMMP2L	CNV	-	chr7:110,662,644–111,562,517 7q31.1	Catalyzes the removal of transit peptides required for the targeting of proteins from the mitochondrial matrix, across the inner membrane, into the inter-membrane space.Known to process the nuclear encoded protein DIABLO	ASD
RELN	CNV	Missense (15) Nonsense (37) Frameshift (33) Splice junction loss (21)	chr7:103,471,381–103,989,658	This gene encodes a large secreted extracellular matrix protein thought to control cell–cell interactions critical for cell positioning and neuronal migration during brain development	Cerebellar hypoplasia Global development delay Seizures Thick cerebral cortex

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
