# Peer review of "Autism Spectrum Disorder Pathogenesis—A Cross-Sectional Literature Review Emphasizing Molecular Aspects"

_ijms, 2024, doi:10.3390/ijms252011283_

Round 1

Reviewer 1 Report (New Reviewer)

Comments and Suggestions for Authors

The manuscript, entitled "Autism spectrum disorder pathogenesis—a cross-sectional literature review emphasizing molecular aspects", offers valuable information about the main streams that focus on ASD etiology.

Few remarks that would improve the quality of the manuscript are enlisted bellow:

I. Introduction is concise and correct. It summarizes the lack of reliable diagnostic biomarkers and specific treatment due to unclear etiology. In addition, the authors emphasize a significant medical problem associated with a surge in both medical and social care costs.

II. Materials and Methods

There is no information on how the review was conducted. The authors should consider adding a methodology section to the manuscript to indicate the type of review i.e., systematic, scoping or narrative? There should be some methodology on how the review was conducted i.e., (i) search terms used, (ii) databases searched and (iii) how the different articles mentioned were identified. A figure (like a PRISMA diagram) might also be useful.

III. The results section is well organized, detailed and logically developed, and the included figures and tables effectively illustrate the information presented, but:

1. It would be more informative, if the table includes an additional column that will emphasize in which part of the gene the particular variation does occur, such as promoter, UTRs or coding sequence and etc.

2. It would be good to include more information on inflammation, mitochondrial dysfunction and oxidative stress as triggers in ASD

IV. The conclusion interprets well the findings of no single pathway being associated with ASD. In addition to these, the need for new experimental evidence for the diagnostic markers of ASD is emphasized.

In conclusion, the topic of the manuscript is very interesting and will stimulate the interest of the reader, because any progress in the understanding of ASD is essential to aid diagnosis and therapy. The manuscript is well prepared and written. Finally, I recommend that the proposed manuscript to be accepted for publication after including the above mentioned points.

Author Response

Dear Reviewer No1

Thank you for your all comments.

  1. There is no information on how the review was conducted. The authors should consider adding a methodology section to the manuscript to indicate the type of review i.e., systematic, scoping or narrative? There should be some methodology on how the review was conducted i.e., (i) search terms used, (ii) databases searched and (iii) how the different articles mentioned were identified. A figure (like a PRISMA diagram) might also be useful.  Answer. Thank you for that comment. All information all placed in Material and method section. We added subsection as 'Searching methodology' .  Moreover, we precised type of review as narrative one. We did not register that review in PRISMA. 

2. The results section is well organized, detailed and logically developed, and the included figures and tables effectively illustrate the information presented, but:

  1. It would be more informative, if the table includes an additional column that will emphasize in which part of the gene the particular variation does occur, such as promoter, UTRs or coding sequence and etc.   Answer - it has been added
  2. It would be good to include more information on inflammation, mitochondrial dysfunction and oxidative stress as triggers . Answer - it has been added

Reviewer 2 Report (New Reviewer)

Comments and Suggestions for Authors

Title: Autism spectrum disorder pathogenesis – a cross-sectional literature review with emphasis to molecular aspect.

In this paper, the authors study autism spectrum disorder (ASD) and present data covering the main streams that focus on the etiology of ASD. Data collection occurred in many countries covering ethnically diverse subjects. Furthermore, the authors aim to show how progress in genetic techniques influences medical explanation.

This paper seems good to us but needs important corrections.

Abstract: 1) remove the phrase caused by the "wrath of God". 2) lacks a conclusion.

Fig. 1 shows only a title without explanation. Expand the legend

The table is very complex and should be divided into at least 2 parts. In addition this table should have a long and clear legend.

-          This paper should be better presented. To make this paper more interesting for the readers of this important journal, the authors, in relation to their data, should talk about inflammation. In this article the authors should briefly review the factors that contribute to the misrecognition of HF ASD and outline improvements in early recognition and intervention. In this regard, below I report 3 interesting articles that should be studied, incorporate their meaning and report them briefly in the discussion and in the list of references.

B. Awidi.  HIDDEN IN PLAIN SIGHT: IMPROVING EARLY RECOGNITION AND INTERVENTIONS IN HIGH-FUNCTIONING AUTISM. European Journal of Neurodegenerative Diseases 2022; 11(2) July-December: 65-69 (www.biolife-publisher.it)

In addition, stress, depression, and dementia are disorders that affect one another and can lead to neurodegeneration. Chronic stress is often linked to chronic inflammatory. This 2 articles should be reported and studied.

1)      R.G. Bellomo. STRESS, DEPRESSION, AND DEMENTIA CONTRIBUTE TO NEURODEGENERATION European Journal of Neurodegenerative Diseases 2023; 12(3) September-December: 86-91. (www.biolife-publisher.it)

2)      G. Ronconi, F. Carinci. AUTISM SPECTRUM DISORDERS (ASDs): NEW RESEARCH AND POSSIBLE NOVEL THERAPIES. European Journal of Neurodegenerative Diseases 2023; 12(2) May-August: 51-56. (www.biolife-publisher.it)

I believe these suggestions are important for improving this paper. Without these corrections the paper cannot be published. So I recommend minor revision.

Comments on the Quality of English Language

Moderate editing of English language required.

Author Response

Dear Reviwer 2.

Thank you for your advice and comments. We appreciate that. 

1) remove the phrase caused by the "wrath of God". 2) lacks a conclusion. Answer - it has been changed

Fig. 1 shows only a title without explanation. Expand the legend. Answer - The Legend is placed under the Figure

The table is very complex and should be divided into at least 2 parts. In addition this table should have a long and clear legend.  Answer - we were confused in this aspect because in Review No1 we got the suggestion to expant the table for additional information.  Please accept this form of table presentation.

This paper should be better presented. To make this paper more interesting for the readers of this important journal, the authors, in relation to their data, should talk about inflammation. In this article the authors should briefly review the factors that contribute to the misrecognition of HF ASD and outline improvements in early recognition and intervention. In this regard, below I report 3 interesting articles that should be studied, incorporate their meaning and report them briefly in the discussion and in the list of references. 

Answer - The additional valuable references have been incorporated to the manuscript.

The manuscript has been edited by MDPI English Service - please find an attached certificate.

This manuscript is a resubmission of an earlier submission. The following is a list of the peer review reports and author responses from that submission.

Round 1

Reviewer 1 Report

Comments and Suggestions for Authors

This review addresses molecular aspects involved in the pathogenesis of ASD. Below I mention some comments that could contribute to improving the presentation of the manuscript:

1.    -     To describe more broadly the inclusion criteria that allowed the selection of the 170,000 participants included in the review. To expand the description of the general characteristics of the 170,000 participants included in the review.

2.     -    Since it is a review it would be advisable to include a figure that illustrates the different molecular aspects involved with the pathogenesis of ASD.

3.      -   Table 1 should mention the bibliographic references.

4.       -  Line 308: mention the method by which the interactions were evaluated.

Author Response

Comments 1.

To describe more broadly the inclusion criteria that allowed the selection of the 170,000 participants included in the review. To expand the description of the general characteristics of the 170,000 participants included in the review.

Answer. According to your comments and also other reviewers this section has been changed.

Comments 2. 

   Since it is a review it would be advisable to include a figure that illustrates the different molecular aspects involved with the pathogenesis of ASD.

Answer - the figure concerning a molecular aspect has been added.

Comments 3   Table 1 should mention the bibliographic references.

Answer - according to your comments and also other reviewers the table 1 has been changed.

Comments 4. Line 308: mention the method by which the interactions were evaluated.

Answer - the sentence in line 308 has been changed.

Please find an attached file with English proofreading certificate.

Reviewer 2 Report

Comments and Suggestions for Authors

Comments on the Quality of English Language

In this review " autism spectrum disorder pathogenesis – a cross-sectional literature review with emphasis to molecular aspect. Abstract mentions that the etiology of autism spectrum disorder (ASD) has not yet been completely elucidated. Through time, multiple attempts have been made to uncover the causes of ASD. Different theories have been proposed, such as that it is caused by “Gods Wrath,” alterations in the gut–brain axis with an emphasis on gut dysbiosis, post-vaccine complications, and genetic or even autoimmune causes.  Authors have just mentioned about post-vaccine complications, and genetic or even autoimmune causes in the current review abstract. But have not discussed anything about post-vaccine complications, and genetic or even autoimmune causes in the current review. In my opinion, the review needs extensive changes in all the sections. There is not much patient data given in the current review. There are less patient studies cited in the review regarding ASD and its risk factors.

Please find my comments about the manuscript.

1.     Title could be changed to “autism spectrum disorder pathogenesis – a cross-sectional literature review emphasizing molecular aspects”.

2.     Please add more key words. Remove key words like asperger syndrome because authors have not written anything about it.

3.     Table 1 presentation of studied mutations in ASD cases according to its function and location is misleading as authors have combined both location of genes and functions. Please make two separate tables to discuss.

4.       Authors mentioned “Diagnosis of ASD by a psychiatrist”. But not a single case has been cited as how a psychiatrist evaluates autistic child. Please add material in support of it as mentioned in line 70..

5.     The standard of English in the manuscript is poor.  Extensive revision of the grammar is needed as most sections are poorly written. There are spelling mistakes at many places in the review. Some sentences are too long. Some of the sentences need to be rephrased. Please correct them.

6.     In the introduction authors write “The diagnosis of ASD is still only symptomatic and is based on ICD-10 criteria” on line 49. What is ICD-10 criteria? Authors have not give any references from this paragraph until the end of introduction section. Please add references.

7.     In the Materials and Methods, authirs mention “This systematic review is in line with PRISMA 2020 guidelines and focuses on…... Is this a systematic review as systematic reviews help a physician or health care provider to make an evidence-based decision in treating his patients. However, this review is least helpful to physicians as there are no studies to support finding from clinic in management of ASD.

8.     In conclusion, authors mention parent-mediated interventions. However, nothing has been discussed in the current review about it.

9.     References are poorly written and haphazard. There is no coherence. Please correct and check the journal format and follow it throughout the reference section.

10.  Also, this review is a brief review. Each section needs additional information as everything is discussed briefly. Clinical decision-making process criteria or pathways is not discussed neither are any molecular signature studies discussed.

Please include and discuss in the present review all these mentioned above points to make this review more focused and beneficial for other researchers who will follow it as a reference.

Author Response

  1. Title could be changed to “autism spectrum disorder pathogenesis – a cross-sectional literature review emphasizing molecular aspects”. Answer- the title has been changed.
  2. Please add more key words. Remove key words like asperger syndrome because authors have not written anything about it. The additional key words have been added. We left Asperger syndrome because it belongs to Autism spectrum disorder.
  3. Table 1 presentation of studied mutations in ASD cases according to its function and location is misleading as authors have combined both location of genes and functions. Please make two separate tables to discuss. Anwer. The table has been rebuilt.
  4. Authors mentioned “Diagnosis of ASD by a psychiatrist”. But not a single case has been cited as how a psychiatrist evaluates autistic child. Please add material in support of it as mentioned in line 70. Answer. This aspect and also ICD-10 have been changed.
  5. The standard of English in the manuscript is poor.  Extensive revision of the grammar is needed as most sections are poorly written. There are spelling mistakes at many places in the review. Some sentences are too long. Some of the sentences need to be rephrased. Please correct them. Answer. The extended proofreading has been made by MDPI English service.
  6. In the introduction authors write “The diagnosis of ASD is still only symptomatic and is based on ICD-10 criteria” on line 49. What is ICD-10 criteria? Authors have not give any references from this paragraph until the end of introduction section. Please add references. Answer. The concern of ICD-10 has been changed.
  7. In the Materials and Methods, authirs mention “This systematic review is in line with PRISMA 2020 guidelines and focuses on…... Is this a systematic review as systematic reviews help a physician or health care provider to make an evidence-based decision in treating his patients. However, this review is least helpful to physicians as there are no studies to support finding from clinic in management of ASD. Answer. The information concerning PRISMA guidelines has been removed because the nature of this review (the narrative type).
  8. In conclusion, authors mention parent-mediated interventions. However, nothing has been discussed in the current review about it. Answer. We added more that.
  9. References are poorly written and haphazard. There is no coherence. Please correct and check the journal format and follow it throughout the reference section. Answer. The references have been checked.
  10. Also, this review is a brief review. Each section needs additional information as everything is discussed briefly. Clinical decision-making process criteria or pathways is not discussed neither are any molecular signature studies discussed. Answer. We added more information in all section.

Please include and discuss in the present review all these mentioned above points to make this review more focused and beneficial for other researchers who will follow it as a reference.

Please find an attached file with English proofreading certificate.

Reviewer 3 Report

Comments and Suggestions for Authors

The current manuscript is a narrative review compiling 69 studies conducted between 1998 and 2020 addressing human patients with autism spectrum disorder (ASD) and largely refraining from studies investigating animal models. Several previous hypotheses about the etiology of ASD (gut-brain axis, environment, stem cells) are discussed and rejected in favor of multi-gene dysfunction as origin of ASD that is now consensus in the scientific community.

This review provides a very general overview. Because the literature after 2020 is not reviewed, the manuscript already needs to be updated - alone more than 2000 reviews were published between 2020 and 2024. Furthermore, many statements are made without providing a reference for the reported data or -only mentioned as an example - reference to Bourgeron is made several times but without indicating the reference (#20).  Ref 54 and 55 are duplicate.

Comments on the Quality of English Language

When updating the manuscript, the style and language should be meticulously revised.

Author Response

The current manuscript is a narrative review compiling 69 studies conducted between 1998 and 2020 addressing human patients with autism spectrum disorder (ASD) and largely refraining from studies investigating animal models. Several previous hypotheses about the etiology of ASD (gut-brain axis, environment, stem cells) are discussed and rejected in favor of multi-gene dysfunction as origin of ASD that is now consensus in the scientific community.

This review provides a very general overview. Because the literature after 2020 is not reviewed, the manuscript already needs to be updated - alone more than 2000 reviews were published between 2020 and 2024. Furthermore, many statements are made without providing a reference for the reported data or -only mentioned as an example - reference to Bourgeron is made several times but without indicating the reference (#20).  Ref 54 and 55 are duplicate.

Answer. We have updated the references. The duplicate references have been removed.

Please find an attached file with English proofreading certificate.

Round 2

Reviewer 2 Report

Comments and Suggestions for Authors

Please reduce the plagiarism as per journal guidelines as your paper still has 26 % plagirism.